# Sensitivity Analysis of Simulation-Based Inference for Galaxy Clustering

**Chirag Modi**
Center for Computational Astrophysics, Flatiron Institute, NY
Center for Computational Mathematics, Flatiron Institute, NY

**Shivam Pandey**
Columbia Astrophysics Laboratory, Columbia University, NY

**Matthew Ho**
CNRS & Sorbonne Université, Institut d'Astrophysique de Paris (IAP), Paris, France

**ChangHoon Hahn**
Department of Astrophysical Sciences, Princeton University, NJ

**Bruno Régaldo-Saint Blancard**
Center for Computational Mathematics, Flatiron Institute, NY

**Benjamin Wandelt**
CNRS & Sorbonne Université, Institut d'Astrophysique de Paris (IAP), Paris, France
Center for Computational Astrophysics, Flatiron Institute, NY

## Abstract

Simulation-based inference (SBI) is a promising approach to leverage high fidelity cosmological simulations and extract information from the non-Gaussian, non-linear scales that cannot be modeled analytically. However, scaling SBI to the next generation of cosmological surveys faces the computational challenge of requiring a large number of accurate simulations over a wide range of cosmologies, while simultaneously encompassing large cosmological volumes at high resolution. This challenge can potentially be mitigated by balancing the accuracy and computational cost for different component models of the simulations while ensuring robust inference. To guide our steps in this, we perform a sensitivity analysis of SBI for galaxy clustering on various main components of the cosmological simulations: gravity model, halo-finder and the galaxy-halo distribution models. We infer cosmological parameters using galaxy power spectrum multipoles (two-point statistics) and the bispectrum monopole (three-point statistics) assuming a galaxy number density expected from current generation of galaxy surveys. We find that SBI is insensitive to changing gravity model between accurate and slow $N$-body simulations and approximate and fast particle mesh simulations. However, changing the methodology of finding the collapsed dark matter structures called halos which galaxies populate can lead to biased cosmological inferences. For models of how galaxies populate these halos, training SBI on more complex model leads to consistent inference for less complex models, but SBI trained on simpler models fails when applied to analyze data from a more complex model.

NeurIPS 2021 AI for Science Workshop.

# 1 Introduction

The three-dimensional distribution of galaxies provides a powerful means to characterize the nature of dark matter and dark energy, to measure sum of the neutrino masses and to test gravity theory on cosmological scales. This has been the focus of various existing, ongoing, and planned galaxy redshift surveys. However, as galaxies are a complex and biased tracer of the underlying matter density field, the complicated process of galaxy formation limits the ease of extracting the cosmological information from the galaxy surveys. While the clustering amplitude of the galaxy density field can be measured to percent-level precision, it cannot straightforwardly be related to the clustering amplitude of the matter density field. Traditional methods of cosmological analysis have also largely been based on using only two- or three-point clustering statistics and analytic models based on perturbation theory (PT) [34, 11, 9]. As a result, these can access only linear and quasi-linear scales and are unable to exploit the full information from galaxy redshift surveys.

Over the last few years, simulation-based inference (SBI), also called likelihood-free inference or implicit-likelihood inference, has emerged as a promising approach to overcome these limitations of traditional analysis [1, 2, 25, 19]. This approach uses high fidelity cosmological simulations (or forward models[1]) to directly model the cosmological observables in full detail. The latest SBI methods combine these simulations with neural density estimation approaches to infer the cosmological parameters efficiently. Using cosmological forward models allows us to use any higher-order summary statistics of the data such as bispectrum, wavelet scattering coefficients, $k$-nearest neighbors or even machine-learnt optimal statistics that can be evaluated in the simulations [*e.g.* 3, 14, 41, 31]. It also enables us to push beyond quasi-linear scales while robustly accounting for observation systematics such as imaging, completeness, fiber-collisions, etc., in our modeling [17, 20]. Meanwhile, since we use neural density estimators, we do not need to assume a Gaussian distribution for the data likelihood but can instead learn the target distributions from the simulations themselves [18]. We refer the readers to [10] for a review on SBI. This method has also recently been applied to analyze survey data for weak lensing in [25] and galaxy clustering data [20].

However, scaling SBI approaches to the next generation of surveys is not straightforward. SBI uses numerical simulations to build a model for analyzing data. Thus the accuracy and robustness of inference with SBI depends to a large extent on- i) the accuracy of the simulators and ii) the number of simulations used to train the SBI procedure. Accounting for both these criterion simultaneously can be challenging. If the underlying simulator does not accurately model the observed data, then the inference is not reliable [8]. This is known as model-misspecification, and the only way to safeguard against it is by using the most accurate simulations for analysis. However, this makes these simulations increasingly computationally expensive and hence for a fixed computational budget, there is a trade-off between the accuracy and the number of these simulations. This challenge is further exacerbated with the increasing volumes of cosmological surveys, and probing observables like emission-line galaxies that increasingly reside in lower mass halos, thus requiring higher resolution simulations. Both of these factors make the simulations more expensive for a given accuracy threshold. To put things in context, the largest simulation suite currently available for training SBI for galaxy clustering (Quijote simulations, Villaescusa-Navarro et al. 42, Hahn et al. 20) consists only of 1000 $(\mathrm{Mpc}/h)^3$ in volume[2], which is smaller than the previous generation surveys that ended a decade ago, and has coarse resolution of 1 $\mathrm{Mpc}/h$. Given the current status, scaling SBI approaches to the scale and fidelity required in the future can be computationally prohibitive and requires strategic planning.

**Motivation** We take first steps towards investigating the simulations requirements for scaling SBI approaches to the next generation of galaxy clustering surveys, and study the sensitivity of SBI to the different components of the forward models used in cosmological simulations. Our goal is to ensure the robustness of inference while balancing the component models to potentially ease the computational requirements. This is motivated by the following observation- the different stages (component models) of simulations have very different computational cost and accuracy. Specifically, for dark-matter only simulations for galaxy clustering, there are three stages in the forward model- i) evolution of dark matter under gravity, ii) finding dark matter halos, and iii) populating these halos with observed galaxies. The gravity evolution is the most computationally expensive part of the

---

[1]In this work, we will use 'simulations' and 'forward models' interchangeably.

[2]Mpc is one mega-parsec, approximately $3 \times 10^6$ light years and $h$ is the dimensionless Hubble parameter that is proportional to the expansion rate of the Universe

simulation, but we are also the most confident in our understanding of the underlying physics. On the other hand, we are the most uncertain about the halo-galaxy connection models, having to infer and marginalize over its parameters during the analysis. This interplay leads us to ask the question- do we need the most accurate models of the gravity evolution if we are uncertain about other components of the model, such as how to populate galaxies in the halos? Does it bias our results if we do not use the most accurate model for all parts of the forward models? A sensitivity analysis of SBI to the different components of the cosmological forward models will answer these questions.

Covering all aspects of this sensitivity analysis is beyond the scope of a single work as the number of cases to investigate increases combinatorially with different components of the forward model, summary statistics, and parameters considered. As a result, here we will focus only on the two traditional summary statistics of galaxy clustering- power spectrum multipoles and bispectrum, but push to smaller scales than the current PT-based analyses [23, 34, 11]. We will focus on only two cosmological parameters- $\Omega_m$ (proportional to the matter density) and $\sigma_8$ (proportional to the matter clustering amplitude), which are well constrained by these statistics. We will consider two component models for each of the aforementioned three stages of these simulations- gravity evolution, halo-finders, and galaxy occupation and study their impact on inference.

We begin in Section 2 by describing the different forward models we will consider for the sensitivity analysis. We describe the simulation data used for each of these models in Section 3 and outline our simulation-based inference methodology in Section 4. Finally we present our results in Section 5 and discuss implications in Section 6.

## 2 Forward Models

In this section, we describe the different models that we will consider for each of the three stages of cosmological simulations. For every stage, we implement two different component models- a simple, often computationally cheap model, and a more complex, often computationally expensive model. Our end-to-end simulations will then consist of all possible combinations of these component models.

### 2.1 Gravity Models

The first step in a cosmological simulation is to evolve dark matter particles under gravity from their initial conditions set at earlier times, to their final distribution at the time of observations. This evolution is generally the most computationally expensive part of the simulations. Here we will consider two different gravity simulations commonly used in cosmology.

**i) $N$-body simulations**   These are the most accurate simulations to evolve cold dark matter (CDM) particles under gravity, for e.g. [16, 38]. $N$-body simulations accurately estimate gravitational forces for particles on all scales, including the particle-particle interactions on the smallest scales at every time-step, and the evolution is simulated with very small (often adaptive) time-stepping for many hundreds of time-steps.

We will use the QUIJOTE $N$-body simulations [42] which simulate $1024^3$ CDM particles in a 1000 Mpc/$h$ box. Each of these simulation requires approximately 5000 CPU hours.

**ii) Particle-mesh simulations**   Particle-mesh (PM) simulations trade-off accuracy for speed as compared to the $N$-body simulations. These estimate the gravitational forces by interpolating CDM particles on a uniform force grid. As a result, these lose information on scales smaller than the grid resolution but are able to solve the Poisson equations using highly efficient fast Fourier transforms. Thus, these simulations are accurate only on the large scales but can be more than $100\times$ cheaper than the $N$-body simulations [e.g. 40, 15]. Recent GPU implementations of PM simulations further increase these computational gains [30, 29].

For this work, we will use FastPM particle-mesh scheme [15]. In each simulation, we evolve $1024^3$ CDM particles on a force grid of $2048^3$ for 10 time-steps. Each simulation required 200 CPU hours, a factor of 10 less than the Quijote simulations.

## 2.2  Halo Model

The next step in cosmology simulations is to find high-density regions called dark matter halos, where the dark matter particles have self-collapsed under gravity. These regions serve as sites for galaxy formation. In this work, we will use two halo-finders commonly used in the community[26].

**i) Friends-of-friends (FoF)**   FoF is a cluster-finding algorithm, where the clusters represent halos in this context. Operationally, FoF finds the clusters in the simulation as follows- if two particles, two clusters, or a particle and a cluster are separated by a distance smaller than a pre-defined distance (linking-length), then they are merged to form a bigger cluster (halo). We use the 3-D FoF halo-finder implemented in NBodykit [21]. By default, this uses a linking-length of $0.2\, l_p$ where $l_p$ is the mean inter-particle distance[3].

**ii) Rockstar**   Rockstar algorithm is a more sophisticated phase-space algorithm for finding halos. We only give an intuition of the algorithm here and refer the reader to the original paper [4] for further details. Briefly, the Rockstar halo finder starts by identifying FoF halos in 3-D position space with a large linking length. It then iteratively refines these clusters using both the positions and velocities of individual CDM particles by pruning those which are inconsistent with expected phase space distribution. These halos are generally considered to be more realistic than FoF halos. Rockstar halo-finder also estimates physical properties of the halo such as its spin, concentration etc., which are not estimated by FoF halos.

## 2.3  Galaxy models

In CDM simulations, dark matter halos need to be populated with galaxies. This is usually done with a statistical framework called the halo-occupation distribution [HOD; 5, 45]. HOD provides a prescription for determining the number of galaxies, as well as their positions and velocities within every halo. The flexibility and accuracy of this framework relates to the number of parameters in the HOD prescription, which need to be inferred and marginalized during analysis. Other approaches to populate galaxies in CDM simulations, such as sub-halo abundance matching (SHAM) and semi-analytic models [37] require additional information from the simulations such as sub-halo distribution and merger trees, but this makes the forward simulations significantly more expensive. Hence here we will focus on using only the following two HOD models.

**i) Zheng07 model**   The standard HOD model [45] assumes that the galaxy occupation depends only on the halo mass, $M_h$. This model has five free HOD parameters which determine the number of central and satellite galaxies: $(\log M_{\min}, \sigma_{\log M}, \log M_0, \log M_1, \alpha)$. Central galaxies are placed at the center of the halos and assigned the velocity same as the halo. Satellite galaxies are placed according to positions and velocities sampled from an NFW profile [32].

**ii) Zheng07ex model**   Our second model extends the standard HOD model by including additional parameters to model assembly, concentration, and velocity biases, leading to a total of 9 free HOD parameters [20]. These are implemented using the decorated HOD prescription of [22]. The assembly bias parameters $(A_c, A_s)$ modify the number of galaxies based on halo concentration. The concentration bias $(\eta_{\mathrm{conc}})$ modifies the positions of satellite galaxies to allow deviation from the NFW profile of their halos. Lastly, the central and satellite velocity biases $(\eta_c, \eta_s)$ re-scale the velocities of central and satellite galaxies with respect to the host halo. This HOD model was used for a recent analysis of a subset of BOSS galaxies in the South Galactic Cap with SBI in [19].

## 2.4  End-to-end forward models

We combine the aforementioned components of our simulations in all possible combinations to generate simulations with different end-to-end forward models to train SBI procedure. However, there are two caveats-

1) Given the two gravity, halo-finding, and HOD models each, we can have a maximum of 8 LH with different forward models. However, in practice, we use only 6 of these as the Rockstar halo-finder

---

[3]In 3-D FoF, all the distances are measured only in the three dimensional position space as opposed to a 6-D phase space.

is not compatible with the PM simulations in its default settings. Due to the missing small-scale forces in PM simulations, the CDM particles are less clustered in phase space and Rockstar with default configuration aggressively prunes these particles resulting in inaccurate halo mass function and clustering. While it may be possible to overcome this by modifying Rockstar, it is out of scope for this work.

2) FoF halo-finder does not estimate halo concentration accurately. Thus in our FoF catalogs, it is instead estimated using analytic mass-concentration formulas from [13]. As a result, in the Zheng07ex model, the assembly bias parameter does not capture bias based on halo assembly but instead only results in a different dependence on halo mass than is included in the standard Zheng07 HOD model. However this caveat should not affect our conclusions.

# 3 Data

In this section, we combine the component models described in the previous section to generate training datasets for simulation-based inference.

## 3.1 Simulations

We use the publicly available simulation suite Quijote for our study here. There are 2000 simulations available at different cosmologies, where we use 1500 simulations for training, 200 for validation and 300 for testing. We generate paired set of 2000 PM simulations at same cosmological parameters and same Gaussian initial conditions as Quijote. Next, we find halos in these simulations. For the $N$-body simulations, we use both Rockstar and FoF. For the PM simulations, we only use FoF for the reasons explained in section 2.4. Finally, we populate each of these three cases, we populate the halo catalogs with galaxies using the 2 HOD models described above. For each halo catalog, we sample 20 different HOD parameter values, resulting in a total of 40,000 galaxy catalogs per forward model. The details of the parameters varied and their prior ranges are detailed in Appendix A.

## 3.2 Summary statistics

In this work, we restrict ourselves to analyzing only the power spectrum multipoles $P_\ell(k)$ for $(\ell = 0, 2, 4)$ and bispectrum monopole $B_0(k_1, k_2, k_3)$. The power spectrum multipoles are measured with fast Fourier transforms using Nbodykit [21] on a $512^3$ mesh. These multipoles are measured in the range $k \in [0.007, 0.5]$ $h$/Mpc, in bins of width $\Delta k = 2\pi/1000 \, h \, \mathrm{Mpc}^{-1}$. This leads to a data vector of 79×3 power spectrum coefficients. During training and testing, we also add to the power spectrum monopole a randomly sampled shot-noise contribution beyond the Poisson shot noise $S_n \sim \mathcal{U}[10^3, 10^4]$, and marginalize over it during inference. This is done to be consistent with previous $P_\ell(k)$ analyses [20, 7, 23, 27]. However, we found that our conclusions remain the same without it.

Bispectrum is measured on a $360^3$ mesh using the `pySpectrum` python package[4], which implements the [36] redshift-space bispectrum estimator. We measure bispectrum in triangle configurations defined by $k_1, k_2, k_3$ bins of width $\Delta k = 3k_f$, where $k_f = 2\pi/(1000 \, h^{-1}\mathrm{Mpc})$ is the fundamental mode. We impose the same scale cut of $k_{\max} = 0.5$ $h$/Mpc as power spectrum, and this leaves us with 1980 triangle configurations. We show the sensitivity of these summary statistics on the gravity model, halo model and HOD model in Appendix B.

# 4 Simulation-based Inference

Next, we outline the details of our simulation-based inference pipeline using the Latin-hypercubes generated in the previous section as the training datasets.

**Methodology:** We have generated a training dataset of $(\theta, \mathbf{x})$ pairs where $\theta$ denotes the cosmology and HOD parameters, and $\mathbf{x}$ denotes the corresponding observations i.e. the power spectrum multipoles and bispectrum. To infer the posterior $p(\theta|\mathbf{x})$, we train a conditional neural density

---

[4]`https://github.com/changhoonhahn/pySpectrum`

estimator $q_\phi(\theta|\mathbf{x})$ with parameters $\phi$ which are fit by maximizing the log-probability of the model parameters conditioned on the data over this training dataset.

**Implementation:** We use the SNPE-C algorithm implemented in `sbi`[5] package to train masked auto-regressive flows (MAF, [33]) as conditional neural density estimators and learn the posterior $q_\phi(\theta|\mathbf{x}) \sim p(\theta|\mathbf{x})$. For robustness, we train 400 networks for each data-statistic by varying hyperparameters corresponding to the width and the number of layers in a single MAF block, number of MAF blocks, learning rate, and the batch size. We use we use `Weights-and-Biases`[6] package for this hyperparmater exploration. After training, we collect 10 neural density estimators with best validation loss and use them as an ensemble *i.e.* we construct a mixture distribution with uniform weighting to approximate the posterior. For posterior inference over a test observation $\mathbf{x}'$, we query the trained ensemble estimator $q_{\phi^*}$ to generate samples from the posterior i.e. $\theta \sim q_{\phi^*}(\theta|\mathbf{x}')$.

**Validation:** To validate that our posteriors are well-specified, we use our trained ensemble to predict the cosmology parameters over the held-out test-dataset from the same forward model as was used for training the ensemble. We use these samples to do coverage tests as described in [39, 20], and verify that all the rank histograms are uniformly distributed within the rank scatter. We will show the coverage plots corresponding to these in the next section. Note that this is a necessary but not a sufficient test to ensure that the posteriors are well calibrated. Furthermore since we use the same forward model for training and testing the SBI procedure in this validation, note that this does not test for model-misspecification.

## 5   Results

We now perform the sensitivity analysis of SBI by looking at the impact of using different component models in training and testing the SBI procedure.

We have generated mock data from six different forward models. We will use these to vary one of the three components (gravity model, halo-finder and HOD model) at a time between the two choices that are described in Section 2, while keeping the other two components fixed. In each case, we will consider inference in the two scenarios- when the test data is generated from the same forward model as the training dataset, and when the test data is generated from another forward model which varies one of the three components. The first scenario validates that our SBI procedure has been trained properly and our posteriors are well calibrated, while the second scenario gauges the impact of model misspecification.

In all cases, we infer the five cosmological and all HOD parameters using power spectrum multipoles and bispectrum. However for the sake of clarity, we present the results only for $\Omega_m$ and $\sigma_8$ which are the two parameters best constrained by these statistics. We present our results in the form of residuals, i.e. the difference between the true and the inferred mean estimate of the parameters over the held out test-dataset, as well as the corresponding posterior standard deviation. Additionally, we also show the coverage plots to verify if the posteriors are well-calibrated, when relevant. In all the figures, we will use blue (and orange) color to show the results for the case when SBI is trained and tested on the same (and different) forward model.

**Gravity models:** We begin by investigating the impact of varying gravity model between the $N$-body and PM simulations. The halo-finder is fixed to FoF since, as discussed earlier, Rockstar halo-finder is incompatible with PM simulations. The HOD model is fixed to 10-parameter Zheng07-ex model.

In Appendix C.1, we show the residuals for SBI trained on both the gravity models when the true data is generated from the $N$-body simulations. We find that the residuals are consistent for the test configurations, *indicating that we are not sensitive to model misspecification in this case*. In Fig. 1, we show the coverage plots indicating that all the posteriors are also well-calibrated and do not under-estimate or over-estimate the posterior widths. Though not shown here, we have checked for consistency that same conclusions hold when the test observations are generated from PM simulations instead of $N$-body simulations, other components kept the same. Overall, these results are promising

---

[5]https://github.com/mackelab/sbi
[6]https://wandb.ai/site

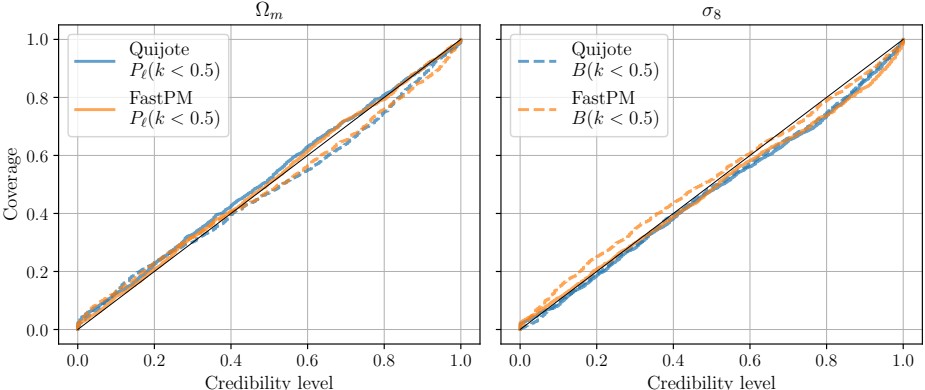

Figure 1: *Gravity models*: Coverage plot corresponding to the posteriors for which residuals are shown in Fig. 5. We use the same color scheme. Power spectrum and bispectrum results are in solid and dashed lines respectively lines. Two columns show the two parameters. Diagonal lines following $y = x$ correspond to perfectly calibrated posteriors.

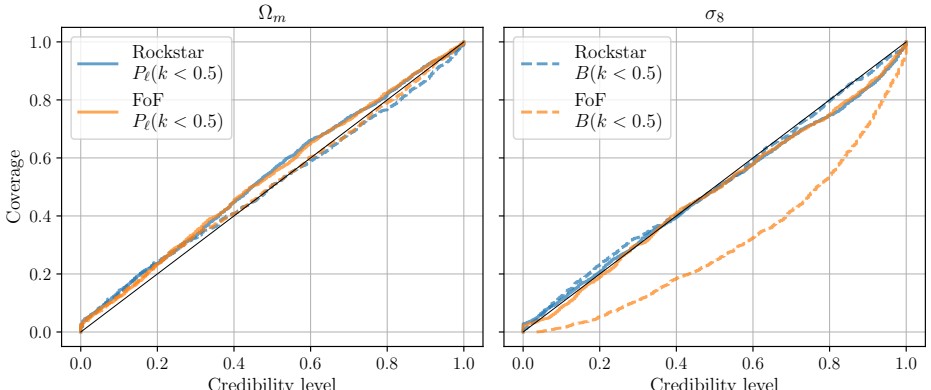

Figure 2: *Halo finders*: Same as Fig.1 but for varying halo-finders. Varying halo-finders for training SBI between Rockstar (blue) and FoF (orange). Test-data is generated with Rockstar halo-finder. The gravity model is fixed to $N$-body and the galaxy model is 10-parameter Zheng07-extended HOD

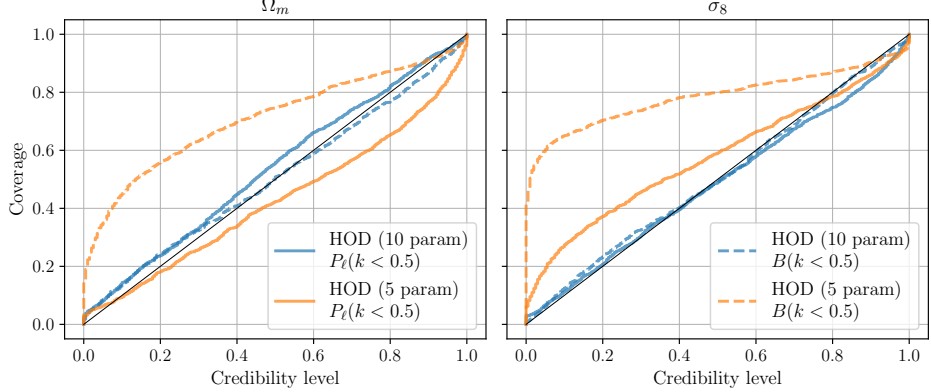

Figure 3: *Galaxy Model II*: Same as Fig.1, but for varying galaxy models. Varying the galaxy model between the 10-parameter (blue) and 5-parameter HOD model (orange). Test-data is generated with 10-parameter HOD. The gravity model is fixed to $N$-body and we use Rockstar halos.

as they indicate that at least for this particular experimental setting, one could generate cheaper training data from PM simulation to infer parameters for the mock data generated from the expensive $N$-body simulations.

**Halo finder models:** Next, we vary the halo-finder in the simulations between FoF and Rockstar. The gravity model is fixed to $N$-body simulations and the HOD model is fixed to 10-parameter Zheng07-extended model. In Appendix C.2 we show the residuals for SBI trained on the two halo finders and applied to test-data generated from the Rockstar halo finder. In Fig. 2 we show the covarage test for the same. In all cases considered, the posterior for $\Omega_m$ seems to be well-calibrated and unbiased. For $\sigma_8$, the posteriors are unbiased when the summary statistic is power spectrum. However when we use bispectrum, SBI trained on Rockstar halos infers well calibrated posteriors for Rockstar data, but the SBI trained on FoF halos consistently under-predicts $\sigma_8$. We observe similar results when the test-data is generated from FoF catalogs. Together, these results clearly indicate that bispectrum statistic is sensitive to differences in halo-finder when inferring $\sigma_8$ and SBI suffers from model-misspecification.

**Galaxy models:** Finally, we change the galaxy occupation model for the simulations between the 5-parameter Zheng07 and 10-parameter Zheng07-extended HOD models. The gravity model is fixed to $N$-body and we use Rockstar halo-finder. In Appendix C.3 we show the results for simpler case when the test-data generated from 5-parameter HOD model. As the 5-parameter HOD model is a subset of the 10-parameter HOD model, SBI trained on both the HOD models gives consistent inference for both the parameters and using either of the summary statistics.

We turn to the more interesting case where the test-data is generated from 10-parameter HOD model. We show the residual plots in Appendix C.3 and coverage plot in Fig. 3. In this case, SBI trained on the correct forward model results in well-calibrated posteriors for both the parameters from both summary statistics. However for SBI trained on the 5-parameter HOD, both power spectrum and bispectrum suffer from model misspecification albeit to different degree. While posterior inferred by power spectrum is still sometimes consistent with the truth, bispectrum almost always leads to incorrect posteriors for both the parameters. This suggests that when trained on a simplistic galaxy occupation model, SBI struggles in doing inference with more complex galaxy models and this is aggravated as the summary statistics used become more informative.

**Summary:** Based on the results of this and the previous section, it is clear that access to accurate galaxy models will likely be the limiting factor in moving forward with all the methods that try to construct models for small scales using cosmological simulations (for e.g. SBI, machine learning and emulator based approaches Yuan et al. 43).

## 6 Discussion and Outlook

We have taken the first steps towards a sensitivity analysis of SBI for galaxy clustering to answer the question- how sensitive are we to different components of our simulations? Studies like this are necessary to scale SBI approaches for the future cosmological surveys, especially as these surveys increase in volume and require higher resolution simulations to model observables.probe observables in lower halo masses. It is becoming increasingly urgent to consider the trade-offs between accuracy and the number of simulations that can be run to generate training datasets.

In this work, we have considered the problem of constraining two cosmological parameters, $\sigma_8$ and $\Omega_m$ from galaxy catalog using power spectrum and bispectrum statistics. We have varied three components of the forward simulations- gravity evolution, halos-finders and galaxy occupation and investigated their impact on inference. We find that inference in the current setup is not sensitive to changing the gravity model between $N$-body and particle mesh simulations. However surprisingly, changing the halo-finder between FoF and Rockstar leads to biased estimate of $\sigma_8$ with bispectrum. For varying galaxy models, SBI results in consistent inference when trained on a 10-parameter HOD model and tested on 5-parameter HOD model, but not the other way round. When trained on 5-parameter HOD and tested on the 10-parameter model, both power spectrum and bispectrum can lead to biased results but the degree of bias for bispectrum is much larger than power spectrum.

As we move towards more powerful statistics like wavelet coefficients, learnt neural summary statistics etc. to extract more information in cosmology, we become increasingly more sensitive to

model misspecification in our simulators. This also serves to guide the new methodologies being developed to accelerate forward simulations [12, 28, 24] i.e. while it is important to report the accuracy of the simulated summary statistics, it is non-trivial to translate these to the expected results of doing inference using these accelerated simulations.

Finally, while we have focused on SBI as a specific tool for inference, the findings are more generally applicalble. Since SBI learns the full likelihood (or the posterior) distribution of the data, it is simply more suited to highlight these issues than the approaches which learn only the mean prediction and assume a Gaussian likelihood. The challenge of robustness is faced by all methods that use simulations for building a data-model (i.e. most machine learning or emulator based frameworks [43]) on small scales where simulations can be unreliable.

## Acknowledgments

This work is supported by the Simons Collaboration on "Learning the Universe". The Flatiron Institute is supported by the Simons Foundation.

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

# A    Simulation details

Our simulated data consists of galaxy catalogs in redshift space at $z = 0.5$. The average number density of galaxies is $\bar{n} = 4 \times 10^{-4}$ $(h/\text{Mpc})^3$ with an average satellite fraction of 20%. We expect similar level of co-moving galaxy number density from the luminous red galaxies (LRG) observed using the DESI survey [46], though our estimate of satellite fraction is approximately 5-10% higher compared to expectations from DESI LRGs [44, 6]. In SBI, we need a training dataset to learn the relationship between the observed data and underlying cosmology parameters over a wide range. Thus, for each of the 6 composite forward models described above, we generate mock galaxy catalogs on a Latin-hypercube (LH) of cosmologies.

As mentioned in the main test, for the $N$-body simulations, we use the publicly available Quijote LH subset [42]. It consists of 2000 simulations varying 5 cosmology parameters over the prior range-

$$\Omega_\text{m} \sim \mathcal{U}[0.1, 0.5], \ \sigma_8 \sim \mathcal{U}[0.6, 1.0], \ \Omega_\text{b} \sim \mathcal{U}[0.03, 0.07], \ n_s \sim \mathcal{U}[0.8, 1.2], \ h \sim \mathcal{U}[0.5, 0.9] \quad (1)$$

The PM simulations are also generated at the same cosmologies as these N-body simulations.

Finally for the HOD parameters, 7 of these are sampled from the following fixed priors to be consistent with previous SBI analysis for galaxy clustering [20]

$$\alpha \sim \mathcal{U}[0.4, 1.0], \qquad \sigma_{\log M} \sim \mathcal{U}[0.3, 0.5], \qquad A_c, A_s \sim \mathcal{N}(0, 0.2) \text{ over } [-1, 1],$$
$$\eta_\text{conc} \sim \mathcal{U}[0.2, 2.0], \quad \eta_c \sim \mathcal{U}[0., 0.7], \quad \eta_s \sim \mathcal{U}[0.2, 2.0].$$

For the 3 mass-based HOD parameters, we define priors that vary with cosmology ($\theta$) as follows

$$\log M_\text{min} \sim \mathcal{U}[\log M_\text{min}^\theta \pm 0.15], \quad \log M_0 \sim \mathcal{U}[\log M_0^\theta \pm 0.2], \quad \log M_1 \sim \mathcal{U}[\log M_1^\theta \pm 0.3]$$

For each cosmology, $M_\text{min}^\theta$, $M_1^\theta$ and $M_2^\theta$ are set to ensure that the number density of generated galaxy catalogs is close to the target number density of $\bar{n} = 4 \times 10^{-4}$. This increases sample efficiency over using the same priors for all the cosmologies, which will need to be quite broad. We estimate $M_\text{min}^\theta$, $M_1^\theta$ and $M_2^\theta$ as follows- given the target number density $\bar{n}$ and average satellite fraction of 0.2, we estimate the average number of centrals $\bar{N}_\text{cen}$. For every cosmology, we use this to determine the halo mass $M_h$ above which the number of halos is the same as $\bar{N}_\text{cen}$ and set $\log M_\text{min}^\theta = \log M_0^\theta = M_h$. With this, we then set $\log M_1^\theta$ to match the average number of satellites assuming a fiducial value of $\alpha = 0.7$

# B    Summary statistics sensitivity

In this section we show how sensitive the power spectrum multipoles and bispectrum monopole are to the variations in simulation inputs.

We compare the summary statistics of our galaxy catalogs for different forward models in Fig. 4. In each column, we vary one component of the simulation at a time and show the ratio of the three summary statistics- monopole, quadrapole and bisepctrum (rows)- for the two different models considered for each component. For consistency, all the lines of the same color have same HOD parameters (except Zheng07 model does not include the 5 assembly bias parameters of the extended model). The largest difference is caused by varying the HOD model between the 5- and 10-parameter models. However even with the same HOD model and parameters, changing gravity models and halo-finder can lead to  10-20% difference in quadrapole and bispectrum.

# C    Residuals between trained SBI and truth

## C.1    Gravity models

In Fig. 5, we show the residuals for SBI trained on both the gravity models when the true data is generated from the $N$-body simulations. For both the summary statistics (rows) and parameters (columns), the residuals are consistent, *indicating that we are not sensitive to model misspecification in this case*. This suggests that marginalizing over the HOD parameters due to the uncertainty in galaxy models indeed outweighs the refinements that happen at small scales with using more accurate gravity models. We note that there is a slight negative slope in the $\sigma_8$ residuals with power spectrum.

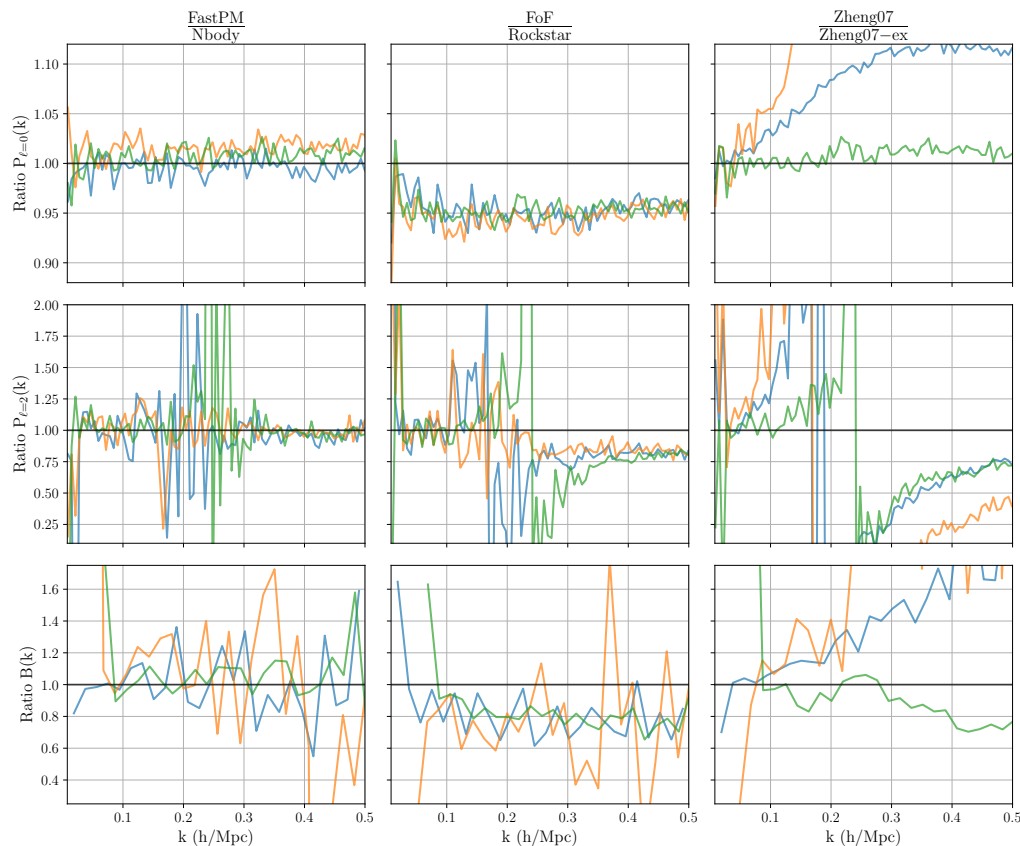

Figure 4: *Comparison of summary statistics for different forward models*: We show the ratio of summary statistics for galaxy catalogs generated by varying one stage of the forward model, as indicated by the title of columns, while keeping the other two stages fixed. The three rows show the ratios for power spectrum monopole (top), quadrapole (middle) and bispectrum (showing equilateral configuration only for clarity, bottom) respectively. The three colors show three different HOD realizations (different parameter values) for the same cosmology. HOD parameters are kept consistent across the columns. The first column shows the ratio for FastPM and $N$-body simulations (with FoF halo-finder and 10-parameter Zheng07-ex HOD model), the second column for simulations with FoF and Rockstar (with $N$-body gravity and Zheng07-ex HOD model), and the third column varies HOD model between 5-parameter Zheng07 and 10-parameter Zheng07-ex model (for $N$-body simulation with Rockstar halo finder).

This effect is consistent with the bounded prior on $\sigma_8$, and would likely go away with a broader prior (relative to the constraint level). However since the same trends exist in both the FastPM and Quijote posteriors, ensuring that the predictive posteriors are consistent, our conclusions regarding model misspecification still hold.

## C.2 Halo-finders

Fig. 6 show the residualsfor SBI trained on the two halo finders and applied to test-data generated from the Rockstar halo finder.

We note that similar analyses were conducted in the robustness tests of SIMBIG[20]. The test sets Test I and Test II of SIMBIGwere designed to assess the sensitivity of a SBI model trained with Rockstar to the choice of the halo finder (FoF and CompaSO). However a direct comparison is not possible since other components of the forward models were varied simultaneously (for Test I, the HOD model was also changed to the 5-parameter Zheng07 HOD model, while for Test II the gravity model was changed to Abacus). These tests were also done only on a single cosmology. Despite this,

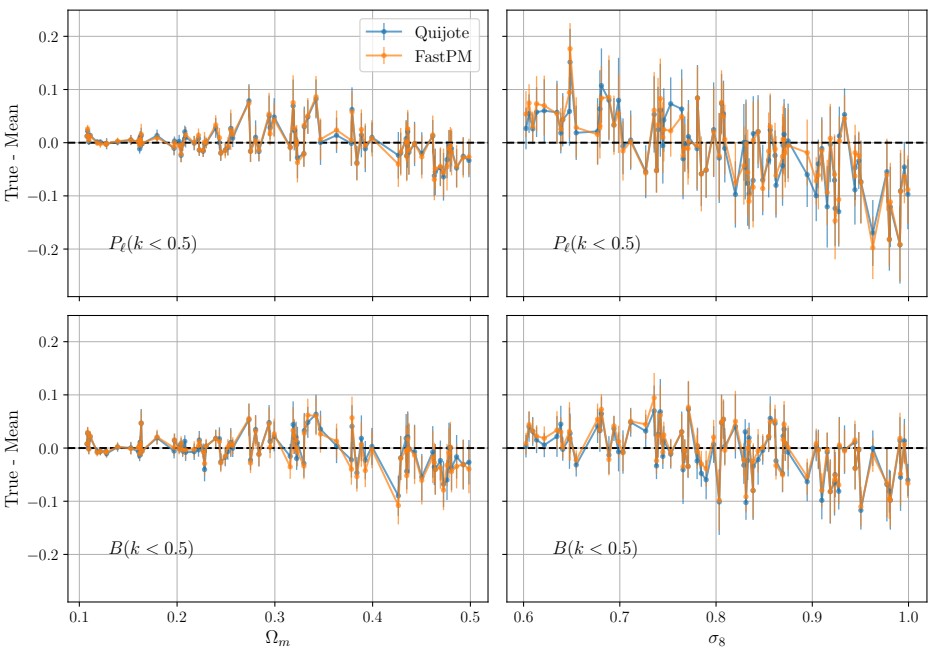

Figure 5: *Gravity models*: Residuals and 1-$\sigma$ posterior width in inferring $\Omega_m$ and $\sigma_8$ (columns) for 100 different simulations with the power spectrum (top row) and bispectrum (bottom row) statistic using SBI. In blue we show results for SBI trained on the correct forward model and in orange with the alternate forward model.

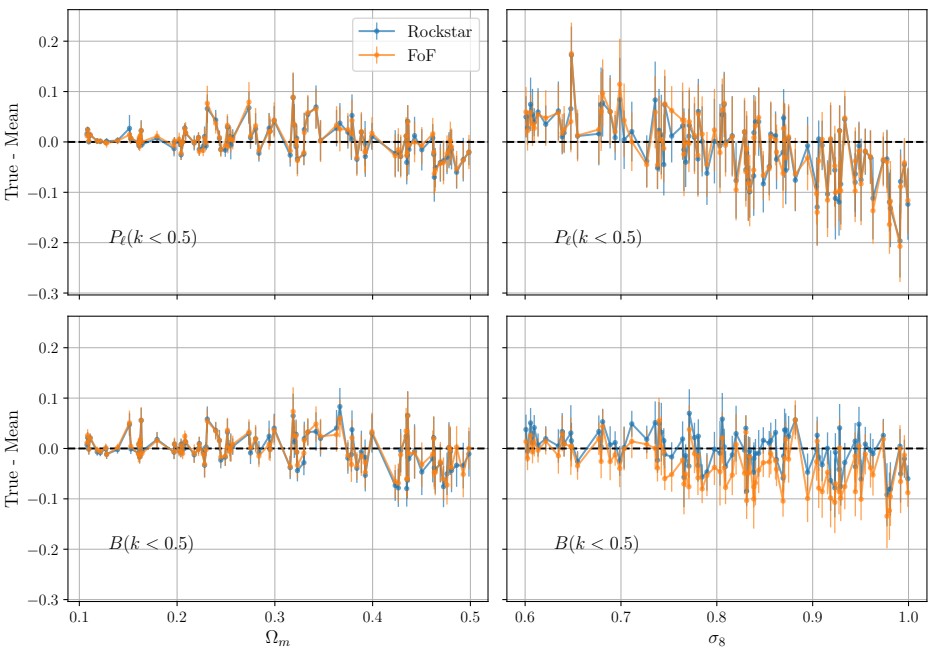

Figure 6: Same as Fig.5 but for varying halo-finders.

similar robustness issues were observed for wavelet scattering statistics in [35] which forced them to use aggressive scale-cuts to mitigate model misspecification.

## C.3  Galaxy models

We begin by considering the test-data generated from 5-parameter HOD model in Fig. 7a and 7b. SBI trained on both the HOD models gives consistent inference for both the parameters and using either of the summary statistics. This is not completely surprising given that the 5-parameter HOD model is a subset of the 10-parameter HOD model, it can simply be generated by setting the assembly bias, concentration and velocity bias parameters to zero.

In the more interesting case in Fig. 8 the test-data is generated from 10-parameter HOD model. We find that bispectrum in particular results in biased and mis-calibrated posteriors.

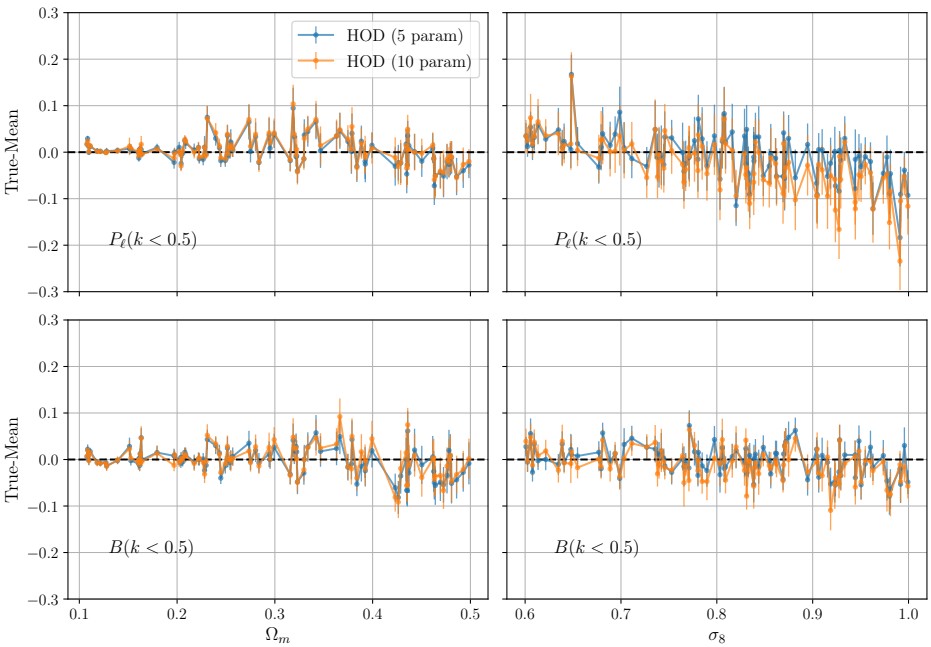

(a) Same as Fig.5 but for varying galaxy models.

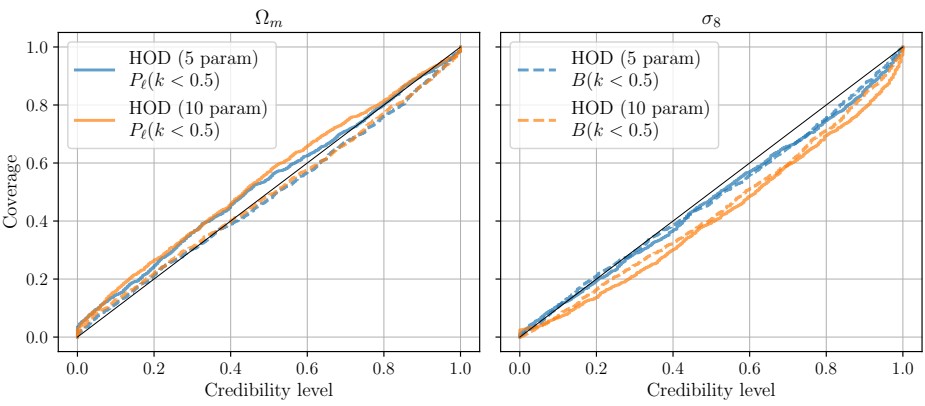

(b) Same as Fig.1 but for varying galaxy models.

Figure 7: *Galaxy Model I*: Varying the galaxy model between the 5-parameter (blue) and 10-parameter HOD model (orange). Test-data is generated with 5-parameter HOD. The gravity model is fixed to $N$-body and we use Rockstar halos.

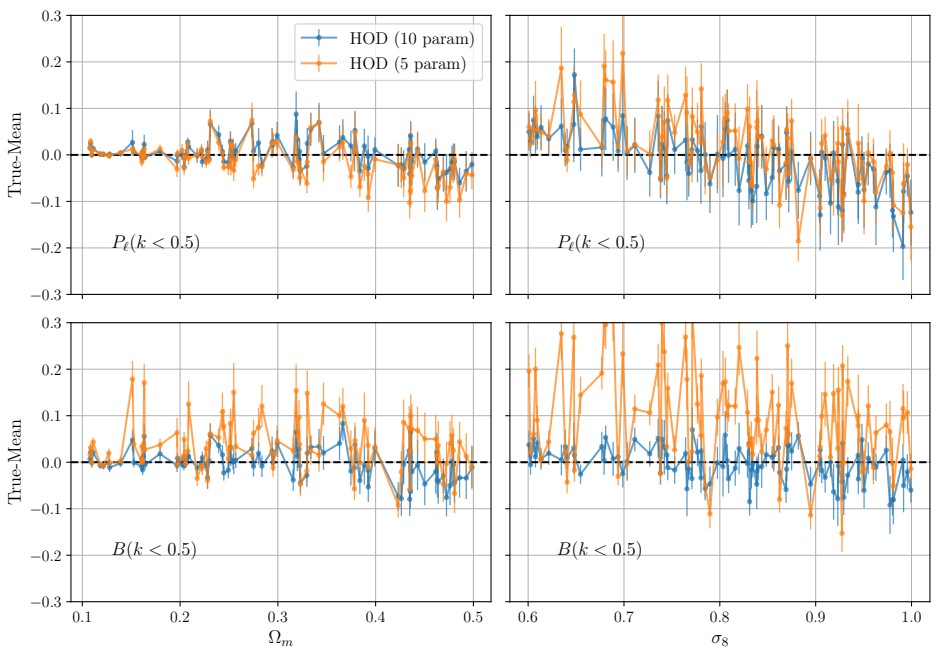

Figure 8: Same as Fig.5, but for varying galaxy models.

