# OpenReview forum: "Sensitivity Analysis of Simulation-Based Inference for Galaxy Clustering"
_NeurIPS.cc/2023/Workshop/AI4Science — NeurIPS2023-AI4Science Poster_

### Official Review · Reviewer_mgB1 · 2023-10-18
**Reviews about Sensitivity Analysis of Simulation-Based Inference for Galaxy Clustering**

**Rating:** 7
**Confidence:** 1

**Review:**

Simulation-based inference (SBI) offers a promising avenue for harnessing high-fidelity cosmological simulations to glean insights from the intricate, non-Gaussian, and non-linear scales that defy analytical modeling. Nevertheless, extending the scope of SBI to meet the demands of the forthcoming generation of cosmological surveys is beset by a substantial computational challenge. This challenge necessitates conducting many precise simulations spanning a wide range of cosmological scenarios while maintaining the capacity to explore large cosmological volumes at high resolutions. One potential strategy to address this challenge entails striking a balance between simulation model accuracy and computational expenditure for various components of the simulations, all while ensuring the reliability of the inferences drawn. To guide our path in this endeavor, we have undertaken a sensitivity analysis of SBI, focusing on galaxy clustering, which involves scrutinizing the core components of cosmological simulations, including the gravity model, halo-finding methodology, and the models governing the distribution of galaxies within halos.


This study aimed to infer cosmological parameters using galaxy power spectrum multipoles (a measure of two-point statistics) and the bispectrum monopole (which captures three-point statistics). Analysis was conducted under the assumption of a galaxy number density that aligns with expectations from the current generation of galaxy surveys. Their findings indicate that SBI exhibits remarkable resilience when confronted with changes in the gravity model. This resilience holds whether the simulations are executed using accurate yet slow N-body simulations or approximate yet faster particle mesh simulations. However, perturbing the methodology for identifying collapsed dark matter structures known as halos, which serve as the cosmic residences for galaxies, can introduce biases into cosmological inferences. When we consider models describing how galaxies occupy these halos, training SBI on more intricate models yields consistent inferences when applied to less complex models. Conversely, SBI trained on simpler models struggle to provide accurate analyses when confronted with data generated from more complex models.

---

### Official Review · Reviewer_QJnh · 2023-10-24
**Comprehensive analysis of different simulation methods**

**Rating:** 8
**Confidence:** 4

**Review:**

**Overview**

Overall, this study takes first step towards investigating the simulations requirements for scaling Simulation-Based Inference (SBI) approaches to the next generation of galaxy clustering surveys.
Three types of models (gravity, halo and galaxy model) are evaluated and the results illustrate that when using simulations for building a data-model on small scales, simulations can be unreliable.

**Advantage**

 1. The research background and motivation of this work are comprehensive and easy to catch up.
 2. Massive experiments and ablation study are conducted, results analysis as well as discussion are convincing.
 3. The findings in the last paragraph are exciting, which can bring inspiration to other researchers in similar research realm.

 **Weakness**

 1. All of the six models used in the paper are existing methods, which weakens the novelty of this work.
 2. Moreover, none of the six models are deep learning related. I wonder if there is other deep learning method that can potentially futher improve performance.
 3. As claimed by authors that "inference in the current setup is not sensitive to changing the gravity model between N-body and particle mesh simulations", what is the reason of this behaviour?

---

### Meta-Review · Area_Chair_zj9w · 2023-10-27

**Recommendation:** Accept (Poster)
**Confidence:** 3

**Metareview:**

This work provides a sensitivity analysis of SBI for galaxy clustering, investigating the requirements for scaling Simulation-Based Inference (SBI) approaches to the next generation of galaxy clustering surveys.

The paper appears scientifically sound and well-presented. Extensive experiments and ablation studies are provided with the conclusion holding promises for exciting applications in related research fields.

The novelty of the work is somewhat limited by the fact that all models used for the analysis already exist. However, the paper still carries interesting insights which are of interest to the community.

In light of this, I follow the positive feedback of the referees and I recommend acceptance for this paper.
Perhaps the authors can look into the 2nd and 3rd bullet points of referee Qjnh and add a few remarks in the paper addressing such concerns.